# THE RELEVANCY METRIC: UNDERSTANDING THE IMPACT OF TRAINING DATA

## ABSTRACT

Deep learning models are central to many critical decision-making processes, making it imperative to gain deeper insights into their behavior to improve performance, transparency, interpretability, and fairness. A key challenge is understanding how training data shapes model predictions on unseen test data. In this paper, we introduce a novel metric, ***Relevancy***, which quantifies the impact of individual training samples on inference predictions. Our proposed metric is calculated by observing the learning dynamics of the model during training, and it is computationally efficient and applicable across a wide range of tasks. We demonstrate that it is between $80\times$ and $100,000\times$ more efficient than existing metrics for capturing the train-test relationship. Using *relevancy*, we enable the identification of coresets — compact datasets that represent the essence of the training distribution. Quantitative evaluations show that coresets selected using our metric outperform state-of-the-art methods by up to $5.2\%$ on CIFAR-100. Additionally, we qualitatively demonstrate how *relevancy* can be extended to assess various training data properties, such as identifying mislabeled samples in widely used datasets like ImageNet, CIFAR-100, and Fashion-MNIST. These examples illustrate just a few of the many potential uses of *relevancy*, highlighting its versatility in promoting more interpretable, efficient, and fair deep learning systems across diverse tasks.

## 1 INTRODUCTION

Deep learning (DL) has achieved significant success in various domains, including classification tasks (Krizhevsky et al., 2009), reinforcement learning (Shakya et al., 2023), diffusion-based image generation (Ho et al., 2020), and text generation (Radford et al., 2019). As tasks grow more complex, both the size and quality of training data play a crucial role in determining model performance. It is widely acknowledged that deep models often overfit, leading to the memorization of training data (Zhang et al., 2017; Arpit et al., 2017). Rather than learning generalized representations, these models may capture specific patterns or memorize individual examples from the training set, which reduces their ability to generalize to unseen data (Brown et al., 2021). Additionally, mislabeled or noisy samples in large-scale datasets further complicate the learning process by encouraging memorization of irrelevant or incorrect patterns, thereby hindering generalization (Northcutt et al., 2021).

To tackle these challenges, it is essential to address two key questions: *"Which subsets of data contribute most to effective generalization?"* and *"How can we identify samples that are memorized?"* Answering these questions requires a deeper understanding of the relationship between training data and model predictions on unseen examples, focusing on balancing memorization and generalization. Recent studies suggest that the learning dynamics of neural networks can provide valuable insights into how models memorize data and generalize to new examples (Toneva et al., 2018; Mangalam & Prabhu, 2019; Jiang et al., 2021; Garg et al., 2024). Building on these insights, we introduce a novel metric, ***Relevancy***, to quantify the impact of individual training samples on generalization.

*Relevancy* measures the influence of a training sample on a model's prediction for any sample of interest. This is achieved by tracking the evolution of the sample losses as the model's weights are updated during training. Specifically, we compute the correlation between the loss trajectories of the two samples over time, using checkpoints saved during training to access these trajectories. When the sample of interest is an unseen data point (e.g., a test sample), this metric enables us to

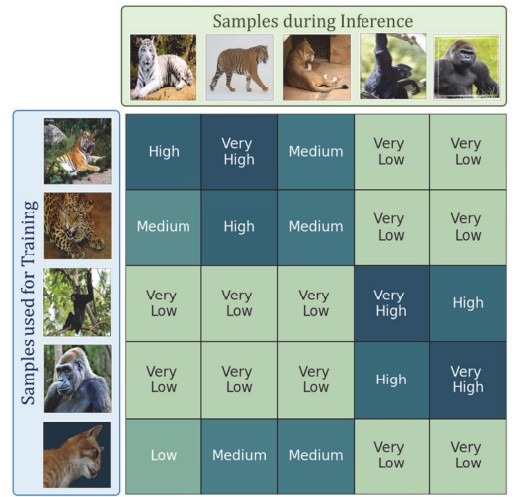
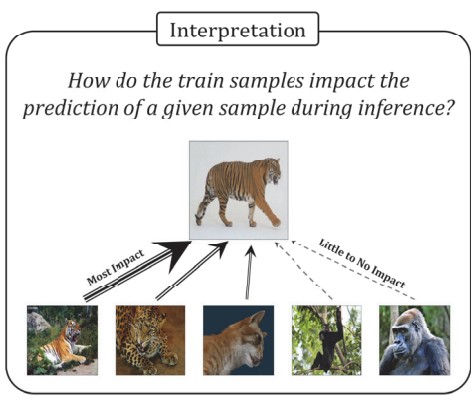

Figure 1: Overview of *relevancy*, showing how individual training samples (rows) affect the predictions of inference samples (columns). Each matrix element represents the *relevancy* score, revealing the contribution of a specific training instance to a given test prediction.

observe how individual training samples contribute to generalization. Conversely, when the samples of interest are from the training set, the metric provides insights into important data properties such as memorization and mislabeled instances (explained in detail in Section 4.2). Figure 1 illustrates the core functionality of *relevancy*, detailing how individual training samples impact inference predictions.

Existing methods for analyzing the train-test relationship, such as influence functions (Weisberg & Cook, 1982; Yeh et al., 2018; Pruthi et al., 2020) and input curvature approaches (Garg & Roy, 2023; Garg et al., 2024; Ravikumar et al., 2024a), provide valuable insights but have limitations in both scope and computational efficiency. Influence functions, for instance, often fail to capture certain types of information (Basu et al., 2021), such as the influence of prototypical samples, due to their bi-modal nature (most scores being approximately 0 or 1) (Lukasik et al., 2024). Additionally, input curvature methods tend to be computationally expensive and do not scale well to large datasets.

In contrast, *relevancy* is designed to be both computationally efficient and scalable. It leverages data collected during training without introducing significant overhead, making it up to $100,000\times$ more computationally efficient than the existing approaches. This makes our method practical for real-world, large-scale applications. By analyzing the correlation over time between the losses of training samples and those of particular samples of interest, whether they are from the test set (unseen examples) or the training set (previously seen examples), *relevancy* offers deeper insights into model behavior.

Our approach addresses the key question: *"Which samples contribute most to model generalization?"* By providing richer information on the relationships between data points, *relevancy* allows for more informed data usage, leading to better model performance and reduced overfitting. Additionally, we demonstrate that *relevancy* can be extended to capture the same information as popular memorization and curvature-based metrics while maintaining superior efficiency and interpretability.

We validate our proposed metric through both quantitative and qualitative evaluations. First, we demonstrate that *relevancy* enables the creation of coresets, compact subsets of the training data that effectively represent the entire dataset. Coresets generated using *relevancy* improve classification accuracy by up to 5.2% on CIFAR-100 (Alex, 2009) with ResNet-18 (He et al., 2016), outperforming state-of-the-art methods. Additionally, we qualitatively show that *relevancy* can be extended to efficiently identify mislabeled samples across various datasets, including ImageNet (Deng et al., 2009), CIFAR-100, and Fashion-MNIST (Xiao et al., 2017), making them powerful tools for data debugging and curation.

In summary, our contributions are as follows:

- **Relevancy:** We introduce *relevancy*, a novel metric that quantifies the influence of individual training samples on model predictions. By tracking changes in loss throughout training, *relevancy* provides fine-grained insights into how specific training samples shape a model's behavior, while offering significant computational efficiency gains over existing methods.
- **Coreset selection:** We showcase the utility of *relevancy* in real-world tasks, particularly in coreset generation, which identifies compact subsets of the original training data that effectively represent the overall data distribution. Our experiments demonstrate that *relevancy*-based coresets outperform state-of-the-art methods by up to $5.2\%$ in classification accuracy on CIFAR-100, while offering interpretable sample selection.
- **Data debugging and interpretability:** We first quantitatively show that we can extend *relevancy* to provide the same insights as existing popular memorization and curvature metrics. We then qualitatively demonstrate that these metrics efficiently identify mislabeled samples across multiple datasets, including ImageNet, CIFAR-100, and Fashion-MNIST, providing actionable insights for data debugging and improving model performance and dataset quality.
- **Scalability and efficiency:** Our method is between $80\times$ to $100,000\times$ more efficient than existing approaches that attempt to measure the relationship between training data and model performance on test data. The computational efficiency makes *relevancy* viable for use in large-scale real-world applications.

These contributions highlight the versatility and practicality of *relevancy*, demonstrating their potential to improve the interpretability, efficiency, and fairness of deep learning systems across a wide range of tasks and can serve as foundational tools for both research and applied machine learning.

## 2 RELATED WORK

Recent research has focused on understanding the influence and memorization of data points in deep learning (DL) models, particularly their implications for model training and generalization. Memorized data refers to samples that a model can only predict accurately when trained on those specific samples, while generalized data leverages knowledge from other samples. Various studies have explored memorization vs. generalization, proposing different definitions and applications, often involving retraining multiple models on different subsets of data or computing expensive Hessian calculations (Zhang et al., 2017; Arpit et al., 2017; Feldman, 2020; Toneva et al., 2018; Brown et al., 2021). Some algorithms, such as those proposed by Garg & Roy (2023); Garg et al. (2024); Ravikumar et al. (2024a;b) leverage input loss curvature to alleviate the computational burden of capturing memorization tendencies but still incur significant computational overheads.

Influence functions, initially introduced by Weisberg & Cook (1982) for regression-based tasks, have been extended to deep learning to measure the influence of training samples on model predictions for unseen data (Koh & Liang, 2017). Methods such as `RandSelect` (Wojnowicz et al., 2016) and Arnoldi iterations (Schioppa et al., 2022) improve computational efficiency by approximating influence scores. Our work is most closely related to `TracIn` (Pruthi et al., 2020), which estimates influence through gradient steps, though it incurs a higher overhead than our proposed metric.

Recent works on training data attributions (TDA), such as Datamodels (Ilyas et al., 2022) and TRAK with its linear datamodeling score (LDS) (Park et al., 2023), evaluate the influence of training data through counterfactual impacts on model outputs. While these approaches provide valuable insights, they often rely on computationally intensive surrogate models or specific assumptions about model linearity. In contrast, our work directly leverages the learning dynamics during training, offering a scalable and assumption-free alternative for understanding train-test relationships.

Our metric also has applications in dataset optimization and mislabel detection. Coreset generation, which selects representative data subsets (Johnson & Guestrin, 2018; Killamsetty et al., 2021a; Paul et al., 2021), addresses computational demands in tasks like Neural Architecture Search (Na et al., 2021; Shim et al., 2021) and continual learning (Aljundi et al., 2019; Borsos et al., 2020). Prior methods include clustering-based `Herding` (Chen et al., 2010), gradient-based `GraNd` (Paul et al., 2021), and bi-level optimization (Killamsetty et al., 2021b). `SloCurves` (Garg & Roy, 2023) highlighted input curvature as a critical factor in identifying impactful samples; however, its computation can be prohibitively costly for large datasets due to repeated Hessian calculations. Mislabel detection is

critical for data quality, and prior work has used pruning and detection tools to identify mislabeled or duplicated samples (Northcutt et al., 2021; Barz & Denzler, 2020). While these methods struggle to scale, our approach, with minimal computational overhead, efficiently handles mislabeled data in large datasets. Our proposed metric, *relevancy*, offers a more computationally efficient approach, achieving state-of-the-art performance with enhanced explainability and minimal overhead.

## 3 THE RELEVANCY METRIC

To ensure that our proposed metric is clearly understood and its significance is effectively conveyed, we now introduce the relevant notations and define our metric with precision and simplicity. Our focus is on capturing nuanced interactions within the training process of neural networks, leading to meaningful insights about sample relationships.

### 3.1 NOTATION

We begin by establishing key notations used throughout this paper for clarity and consistency. Random variables will be represented in bold (e.g., $\mathbf{V}$), with scalar instances denoted by lowercase letters (e.g., $v$), and vectors by arrowed letters (e.g., $\vec{v}$).

In this study, we consider a supervised learning task in which a randomized algorithm $\mathcal{A}$, such as Stochastic Gradient Descent (SGD) (Bottou, 2010), is used to learn a mapping $f : \vec{x} \to y$, where $\vec{x} \sim \mathbf{X} \in \mathbb{R}^d$ and $y \sim \mathbf{Y} \in \mathbb{R}$. The algorithm $\mathcal{A}$ is used to train a model on a dataset $S \sim \mathbf{Z}^m$, where $\mathbf{Z} = \mathbf{X} \times \mathbf{Y}$ represents a joint distribution of input-output pairs. $S$ contains $m$ samples: $S = [\vec{z}_1, \cdots, \vec{z}_m]$, where each sample $\vec{z}_i = (\vec{x}_i, y_i) \sim \mathbf{Z}$. After $t$ epochs of training, algorithm $\mathcal{A}$ produces a model $h_S^{\phi,t}$, where $\phi \sim \mathbf{\Phi}$ represents the randomness in the learning process. During training, the model's performance is optimized using a loss function $\ell$, evaluated on each sample $\vec{z}_i$. The loss at epoch $t$ for sample $\vec{z}_i$ is denoted as $\ell(h_S^{\phi,t}, \vec{z}_i)$.

### 3.2 OVERVIEW

We propose a novel metric, *relevancy*, denoted as $\mathtt{Rel}$, to quantify the impact of a training sample $\vec{z}_i$ on a sample of interest $\vec{z}_j$ (which can be in the training set $S$ or be an unseen sample). This metric reflects the relationship between the evolving learning patterns of the samples within neural networks throughout the training process. For a randomized algorithm $\mathcal{A}$ trained on a dataset $S \sim \mathbf{Z}^m$ over $T$ epochs, the *relevancy* score is defined as the Pearson correlation (Pearson, 1895) between the loss trajectories of $\vec{z}_i$ and $\vec{z}_j$ across the training epochs.

$$\mathtt{Rel}(\mathcal{A}, S, i, j) \coloneqq \mathtt{corr}\left( \ell(h_S^{\phi,t}, \vec{z}_i)\big|_{t=0}^T, \ell(h_S^{\phi,t}, \vec{z}_j)\big|_{t=0}^T \right) \tag{1}$$

This measures how closely the model's loss on the sample of interest $\vec{z}_j$ correlates with the loss incurred on training sample $\vec{z}_i$. The loss trajectory of unseen samples of interest can be extracted by using checkpoints of the model saved during training. More formally, it is computed as:

$$\mathtt{Rel}(\mathcal{A}, S, i, j) = \frac{\sum_{t=1}^T \left( \ell(h_S^{\phi,t}, \vec{z}_i) - \hat{\ell}(h_S^{\phi}, \vec{z}_i) \right) \cdot \left( \ell(h_S^{\phi,t}, \vec{z}_j) - \hat{\ell}(h_S^{\phi}, \vec{z}_j) \right)}{\sqrt{\sum_{t=1}^T \left( \ell(h_S^{\phi,t}, \vec{z}_i) - \hat{\ell}(h_S^{\phi}, \vec{z}_i) \right)^2} \cdot \sqrt{\sum_{t=1}^T \left( \ell(h_S^{\phi,t}, \vec{z}_j) - \hat{\ell}(h_S^{\phi}, \vec{z}_j) \right)^2}}$$

where $\hat{\ell}(h_S^{\phi}, \vec{z}_i)$ represents the mean loss of sample $\vec{z}_i$ over all training epochs. This metric evaluates the alignment between the losses of $\vec{z}_i$ and $\vec{z}_j$ during training, reflecting how learning $\vec{z}_i$ impacts the prediction of $\vec{z}_j$.

A positive *relevancy* value indicates that the training sample $\vec{z}_i$ and the sample $\vec{z}_j$ exhibit similar learning dynamics. If the loss for both samples decreases together during training, it suggests that $\vec{z}_i$ positively influences the model's performance on $\vec{z}_j$, implying that their features are aligned and learning from $\vec{z}_i$ helps in predicting $\vec{z}_j$ more confidently. Figure 2 (top) shows this scenario, where visually similar samples have correlated loss reductions.

Conversely, a negative *relevancy* value indicates opposing learning dynamics, where minimizing the loss on $\vec{z}_i$ increases the loss on $\vec{z}_j$, suggesting dissimilar or conflicting features between the

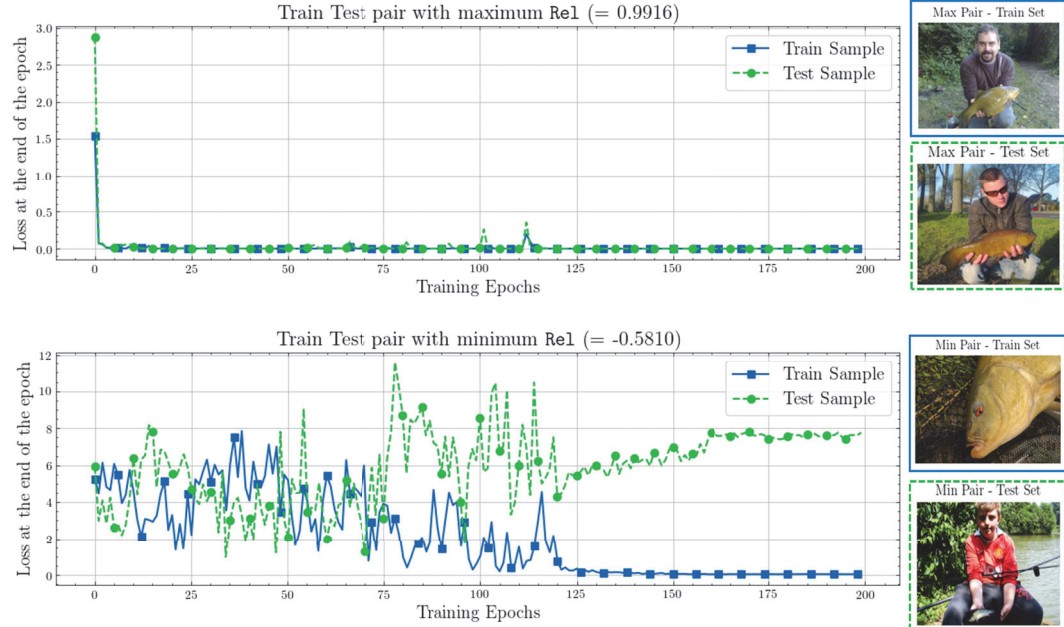

Figure 2: Visualization of the *relevancy* metric for two pairs of samples from Class 0 in ImageNet (Deng et al., 2009). The samples of interest ($\vec{z}_j$) are from the test dataset. The top plot shows the pair with the maximum `Rel` value, where both samples have similar loss trajectories, indicating positive correlation. The bottom plot shows the pair with the minimum `Rel` value, with opposing loss trajectories, indicating a negative correlation. Corresponding images highlight the visual similarity or dissimilarity between the samples.

samples. Figure 2 (bottom) illustrates this with visually dissimilar samples whose losses diverge during training.

These patterns, positive correlation for similar samples and negative correlation for dissimilar ones, make *relevancy* a powerful tool for understanding inter-sample influence during learning. A high *relevancy* score reflects a positive impact, while a negative score captures conflicting signals.

Figure 3 shows examples of the training samples with the highest and lowest *relevancy* scores (from the same class) for randomly selected unseen samples of interest (test samples) in two popular computer vision datasets. Additional visualizations are provided in Appendix B.

## 4 COMPARISON TO POPULAR METRICS

In this section, we compare our proposed metrics to popular methods used for evaluating training sample influence and identifying memorized samples, demonstrating that our approach is both comprehensive and computationally efficient.

### 4.1 INFLUENCE FUNCTIONS

Influence functions are widely used to quantify the impact of individual training samples on model predictions. One of the most recognized metrics in this class is `infl`, introduced by Feldman & Zhang (2020), which is particularly notable due to precomputed influence scores available for datasets like ImageNet. However, `infl` has significant limitations in capturing the influence of highly representative (prototypical) samples, as highlighted by Lukasik et al. (2024).

Figure 4a shows a heatmap of the 2D `infl` matrix (train sample ID × test sample ID) for class 0 in the ImageNet dataset. Most values are zero, with fewer than 33% of train-test pairs having non-zero `infl` values, occurring for only 16 out of 50 test samples. This occurs because `infl` measures the change in prediction probability for a test sample $\vec{z}_j$ when inferred by a model trained with the

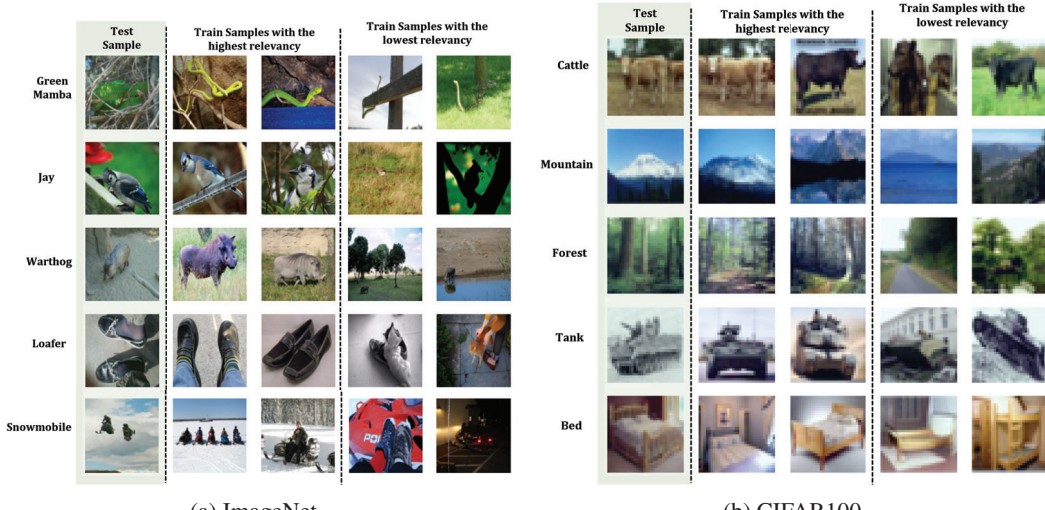

(a) ImageNet         (b) CIFAR100

Figure 3: Examples of randomly selected unseen samples of interest (selected from the test set) and the corresponding training samples (from the same class) with the highest and lowest `Rel` scores in popular computer vision datasets.

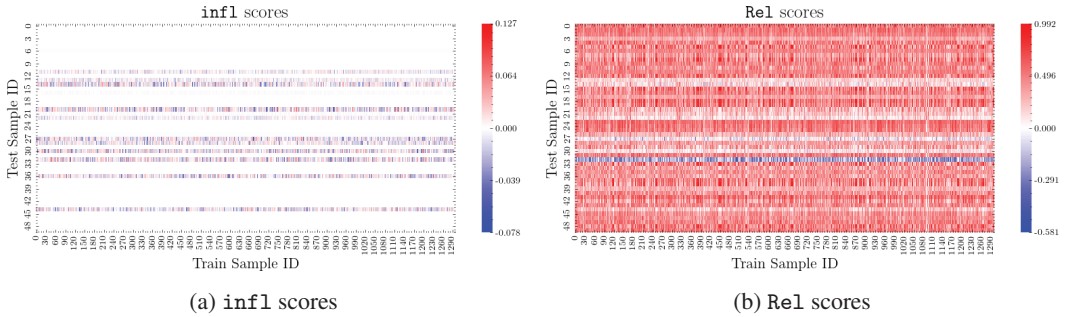

(a) `infl` scores         (b) `Rel` scores

Figure 4: Comparison of `infl` and `Rel` scores for class $0$ of the ImageNet dataset. Note that all test samples have at least one non-zero `Rel` score with a training sample, indicating which training samples impact which test samples. In contrast, most `infl` scores are zero, revealing limited insight into the influence of the training samples. (Best viewed in color).

training sample $\vec{z}_i$, compared to a model trained without $\vec{z}_i$. The full definition is provided in Appendix F. "Prototypical" test samples, like $j = 0$, share features with many training samples and are less affected by the removal of a **single** sample, leading to limited insight from `infl` scores. In contrast, `Rel` provides a more comprehensive view by capturing the influence between every train-test pair, even for prototypical samples. Figure 4b illustrates this, where `Rel` scores reveal meaningful relationships across all test samples, offering a richer understanding of influence dynamics and highlighting a broader range of impactful interactions compared to `infl`.

`TracIn` (Pruthi et al., 2020) is another popular metric for measuring train-test relationships. This requires gradient calculations for both train and test samples at each epoch, making it computationally expensive. In contrast, *relevancy* utilizes readily available loss values, making it computationally efficient. This is further discussed in Section 4.3.

## 4.2 MEMORIZATION AND CURVATURE

Memorization occurs when a model over-relies on specific training examples instead of generalizing to unseen data. While some memorization is necessary, it is crucial to identify overly memorized samples to understand model behavior. The most common memorization metric, `mem`, introduced by Feldman (2020), calculates memorization similarly to `infl` by retraining models with excluded

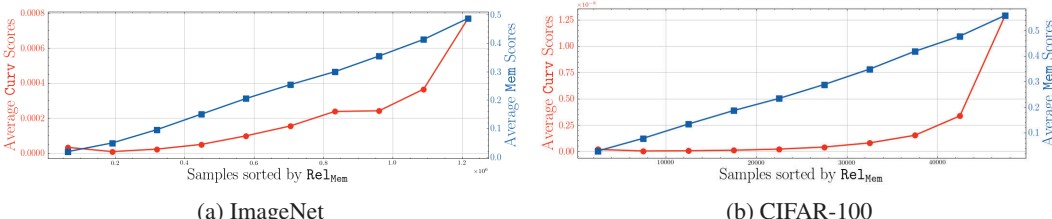

(a) ImageNet                                      (b) CIFAR-100

Figure 5: Comparison of Rel_Mem scores of ImageNet and CIFAR-100 samples to their corresponding mem and Curv scores. Samples were sorted by Rel_Mem, grouped into bins, and the average mem and Curv scores for each bin were plotted. Samples with higher mem or Curv scores tend to have higher Rel_Mem scores, while those with lower scores generally correspond to lower Rel_Mem scores.

samples. However, retraining models for every sample excluded is computationally expensive. An alternative, Curv (Garg et al., 2024), estimates memorization using input curvature, but this requires approximating the Hessian, which is also inefficient. Although, practical implementations of Curv utilize Hutchinson's Trace Estimator (Hutchinson, 1989) for calculating a proxy for the Hessian, it still incurs significant computational overhead.

To identify memorized samples in a computationally efficient manner, we must identify how unique a sample is (i.e., *"Does it possess features that can be generalized?"*) and how much other training samples impact the prediction of this sample (i.e., *"If this sample were not in the training set $S$, would the other training samples assist in its accurate prediction?"*). The latter can be estimated by observing the average *relevancy* of other training samples on the given sample.

To quantify uniqueness, we introduce the measure atypicality (Atyp). We define Atyp as the ratio of the sample's mean loss to the average mean loss across all samples of the same class:

$$\texttt{Atyp}(\mathcal{A}, S, i) := \frac{\hat{\ell}(h_S^\phi, \vec{z}_i)}{\frac{1}{c} \cdot \sum_{i'=0}^c \hat{\ell}(h_S^\phi, \vec{z}_{i'})}$$

where $c$ is the number of training samples from the same class as $\vec{z}_i$. Atypicality captures how much a sample deviates from the average behavior of its class. Samples that closely follow the learning patterns of their class are considered "prototypical" and have Atyp values close to 1, while more difficult or unique samples ("atypical") exhibit Atyp values greater than 1. Visual examples of prototypical and atypical samples are provided in Appendix B.

Utilizing the above two factors (average *relevancy* and atypicality), we can then extend *relevancy* to estimate the degree of memorization for a training sample $\vec{z}_i$.

For a randomized algorithm $\mathcal{A}$ trained on the dataset $S \sim \mathbf{Z}$ for $T$ epochs, we define the memorization score (Rel_Mem) of a sample $\vec{z}_i$ as the average *relevancy* score of all other training samples $\vec{z}_{i'}$ ($i' = 0, 1, 2, \cdots, c; \quad i' \neq i$) of the same class, scaled by the sample's atypicality $\texttt{Atyp}(\mathcal{A}, S, i)$. More formally,

$$\texttt{Rel}_{\texttt{Mem}}(\mathcal{A}, S, i) := \frac{1}{c-1} \left( \sum_{\substack{i'=0 \\ i' \neq i}}^c \texttt{Rel}(\mathcal{A}, S, i, i') \right) \cdot \texttt{Atyp}(\mathcal{A}, S, i) \tag{2}$$

A high Rel_Mem score indicates that the model is likely memorizing the sample, as it is unique and receives little influence from other training examples. A low Rel_Mem score suggests the sample is more generalizable, as it is better supported by other samples in the dataset.

Figure 5 compares the Rel_Mem scores of ImageNet and CIFAR-100 samples with their respective mem and Curv scores. First, we sorted the samples by their Rel_Mem values and grouped them into bins. For each bin, we calculated the average mem and Curv scores and plotted these averages. As shown in the plot, samples with with higher mem or Curv scores tend to have higher Rel_Mem scores, while those with lower mem and Curv scores generally correspond to lower Rel_Mem scores. This suggests that samples flagged as memorized or mislabeled by Rel_Mem (ones with higher scores) tend to overlap with those identified by mem and Curv.

Table 1: Comparison of the computational overhead of popular relational and training data metrics

| Metric | Additional Compute | Example Scenario | |
|---|---|---|---|
| | | Number of FLOPS | Comp. Overhead |
| infl | $(T \cdot s \cdot m \cdot 3f + m' \cdot f) \cdot r$ | $8.807 \times 10^{20}$ | **107640.25×** |
| TracIn | $T \cdot (m' + m) \cdot 3f$ | $6.535 \times 10^{17}$ | **79.87×** |
| mem | $(T \cdot s \cdot m \cdot 3f) \cdot r$ | $8.805 \times 10^{20}$ | **107618.03×** |
| Curv | $T \cdot (1 + r) \cdot m \cdot 3f$ | $6.918 \times 10^{18}$ | **845.57×** |
| **Ours** (Rel) | $\boldsymbol{T \cdot m' \cdot f}$ | $\mathbf{8.182 \times 10^{15}}$ | **1×** |

## 4.3 COMPUTATIONAL OVERHEAD OF THE PROPOSED METRICS

The computational overhead of *relevancy* is minimal. The loss trajectories of training samples are available as a by-product of training and can be obtained directly. To compute the *relevancy* for training samples on unseen samples of interest, we only need to obtain the loss trajectories for the unseen samples. Metrics such as infl (Feldman & Zhang, 2020), TracIn (Pruthi et al., 2020), mem (Feldman, 2020), and Curv (Garg et al., 2024) discussed in Sections 4.1 and 4.2 require expensive gradient calculations or retraining multiple models. Table 1 compares the computational costs of *relevancy* with these metrics, highlighting its efficiency. A detailed explanation of how the computation of these metrics is calculated is provided in Appendix F.

**Notation and Setup:** We denote the number of training samples as $m$ ($S \sim \mathbf{Z}^m$) and the number of samples of interest as $m'$. The model is trained for $T$ epochs, and for certain methods, there are hyperparameters (denoted as $r$) that further influence the computational cost. The computational complexity of each method is expressed in terms of floating point operations per second (FLOPS), denoted as $f$, which represent the cost of a single forward pass for a model with $p$ parameters. The cost of a backward pass is approximately $2f$ FLOPs. For infl, TracIn, and Rel, we consider the scenario of estimating the impact of every training sample on every test sample. For mem and Curv, we consider the scenario of estimating the memorization of every training sample. $\text{Rel}_{\text{Mem}}$ is not included in this table as it only adds an insignificant overhead (due to the calculation of correlation and atypicality) as the loss values of training samples are readily available.

**Example with Imagenet and ResNet-18:** To further illustrate the computational efficiency of *relevancy*, we provide an example using the ImageNet dataset ($m = 1,281,167$ training samples and $m' = 50,000$ test samples of interest) and a ResNet-18 model with $p = 11,689,128$ parameters and $f = 1,818,228,160$ FLOPS per forward pass. We considered the training recipe provided by the popular Bearpaw library (Yang, 2017), where $T = 90$. For the Curv metric, we used $r = 10$ as recommended by the authors. Additionally, we trained $r = 2000$ models using training subsets of size $s = 0.7$ to calculate the expected values for mem, and infl, as suggested by the authors.

## 5 CORESET GENERATION

Coreset generation is crucial for reducing the size of training datasets while retaining the most informative samples, enabling efficient training, and maintaining high model performance. Coresets are particularly valuable in real-time applications with limited resources or when rapid model updates are necessary. They also contribute to model interpretability by focusing on the most influential data points.

To construct coresets, we select training samples based on their average *relevancy* scores relative to the validation set, denoted as $\text{Rel}_{\text{Avg}}$. This metric identifies the samples that have the greatest impact on model generalization:

$$\text{Rel}_{\text{Avg}}(\mathcal{A}, S, i) \coloneqq \frac{1}{m_v} \left( \sum_{j=0}^{m_v} \text{Rel}(\mathcal{A}, S, i, j) \right) \quad (3)$$

where $m_v$ represents the number of available validation samples. This approach effectively reduces training time while preserving model performance by focusing on high-impact samples.

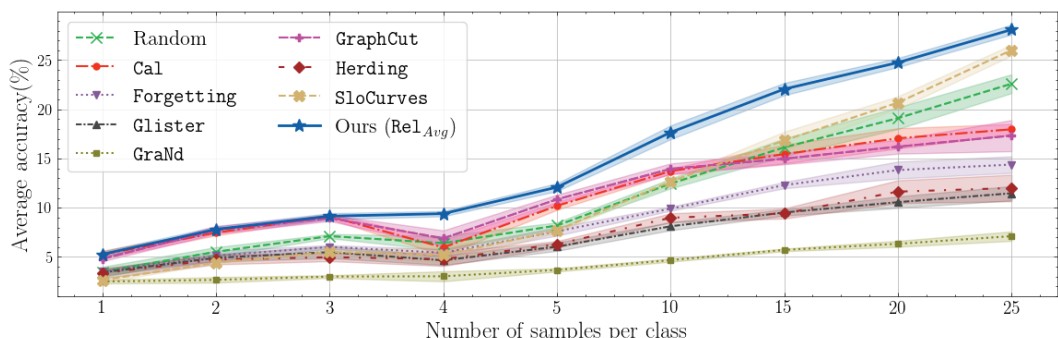

Figure 6: Comparison of various coreset generation techniques for CIFAR-100 using ResNet-18. Each line represents the mean accuracy over five runs (with different random seeds), and the shaded regions show the standard deviation. Additional results for various coreset sizes are provided in Appendix C.

**Experimental Setup:** We conducted experiments on the CIFAR-100 dataset, which contains $50,000$ training samples and $10,000$ test samples distributed across 100 classes. We evaluated coresets of varying sizes (ranging from $0.2\%$ to $10\%$ of the full dataset) using a ResNet-18 architecture. Our method was compared against several state-of-the-art coreset generation techniques, including `Glister` (Killamsetty et al., 2021b), `Forgetting` (Toneva et al., 2018), `GraphCut` (Iyer et al., 2021), `Cal` (Margatina et al., 2021), `GraNd` (Paul et al., 2021), `Herding` (Chen et al., 2010), and `SloCurves` (Garg & Roy, 2023), using implementations from the Deepcore library (Guo et al., 2022). All methods were trained with identical hyperparameters and no additional fine-tuning, ensuring a fair comparison. For methods requiring training for coreset selection, we ran the training for 40 epochs to ensure convergence. Further details of the setup are provided in Appendix C. To ensure a fair comparison, no additional techniques such as regularization were applied during learning (suggested in `SloCurves`), as this can be applied uniformly across all methods.

**Results:** As shown in Figure 6, our coreset generation method consistently outperforms other techniques while being computationally efficient. Even when the coreset size exceeds $5\%$ of the dataset, our approach remains competitive with others. Additional results for various coreset sizes are provided in Appendix C.

**Takeaway:** Coresets generated using the average *relevancy* scores prioritize samples with the highest potential for generalization, leading to superior performance over existing methods. Moreover, this approach is computationally efficient, requiring less computation than other state-of-the-art techniques.

# 6 DISCUSSION AND QUALITATIVE ANALYSIS

## 6.1 IDENTIFYING MISLABELED SAMPLES

Mislabeled samples often compel models to memorize incorrect labels, which can be identified by high $Rel_{Mem}$ scores. Figure 7 shows examples of some of the mislabeled samples that were detected using $Rel_{Mem}$ from popular datasets, emphasizing the value of *relevancy* and `Atyp` for improving data quality by identifying problematic examples. It is worth mentioning that the samples that are identified as mislabeled samples by `Curv` and `mem` metrics, are also identified by $Rel_{Mem}$. This is becuase, all three metrics exhibit similar trends as explained in Section 4.2.

## 6.2 CONSISTENCY ACROSS ARCHITECTURES AND RANDOM SEEDS

One of the strengths of *relevancy* is its consistency across different architectures and random seeds. Despite variations in model design and initialization, the relative *relevancy* scores between training and test samples remain highly stable, ensuring reliable insights into sample influence. To evaluate this, we experimented with different architectures, including various ResNet sizes, VGG-19 (Simonyan & Zisserman, 2014), DenseNet (Huang et al., 2017), and AlexNet (Krizhevsky et al.,

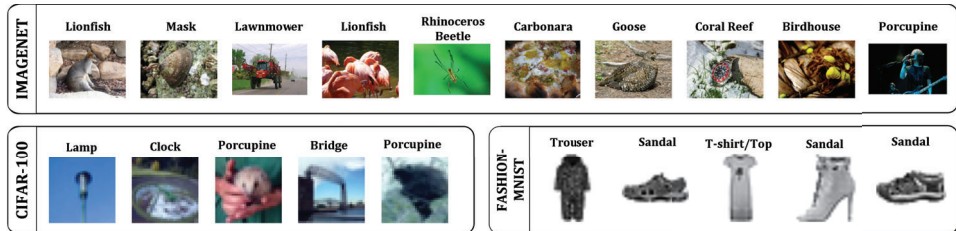

Figure 7: Examples of mislabeled training samples (randomly selected) across classes of popular computer vision datasets. The (incorrect) labels provided by the dataset are indicated above each sample.

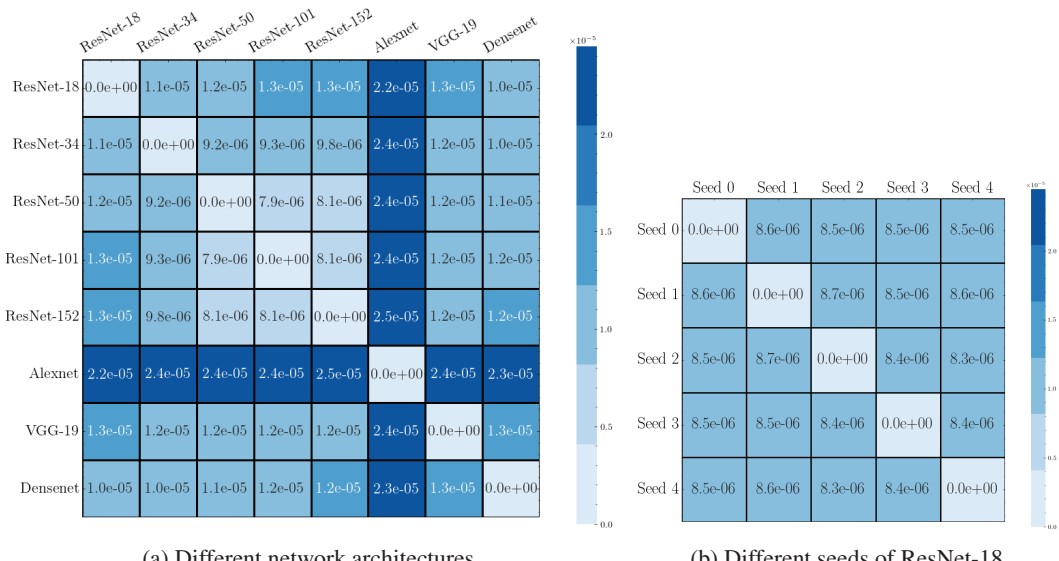

(a) Different network architectures          (b) Different seeds of ResNet-18

Figure 8: Confusion matrices illustrating the mean absolute error (MAE) of `Rel` scores for train-test sample pairs. (a) MAE across architectures trained on CIFAR100 (b) MAE across seeds of ResNet-18 trained on CIFAR100.

2012), as well as different random seeds for ResNet-18 on the CIFAR-100 dataset. The mean absolute error (MAE) between *relevancy* scores across models and seeds was computed for all train-test pairs, as shown in Figure 8. It demonstrates that *relevancy* scores remain highly consistent across both architectures and random seeds, with MAE as low as $10^{-5}$. This makes it a reliable setup agnostic tool for applications like coreset generation.

## 7 CONCLUSION

In this paper, we introduced ***Relevancy***, a novel and computationally efficient metric for quantifying the influence of individual training samples on model predictions, focusing on their impact on unseen test data. By tracking learning dynamics during training, *relevancy* reduces overhead by up to $100,000\times$ compared to existing approaches. We demonstrated its effectiveness in generating coresets, which improved classification accuracy by up to $5.2\%$ on CIFAR-100, outperforming state-of-the-art methods. Additionally, we showed that we can extend *relevancy* to capture the same information as existing popular and computationally expensive memorization and curvature metrics. We qualitatively showed that this extension of *relevancy* can efficiently detect mislabeled samples across datasets such as CIFAR-100, ImageNet, and Fashion-MNIST. These results highlight the versatility and efficiency of our approach, making it applicable to various deep learning tasks where understanding data influence is crucial. As deep learning models are increasingly used in critical decision-making systems, metrics like *relevancy* are essential for ensuring models are both high-performing and interpretable. Future work will focus on extending these metrics to different architectures and tasks, advancing their role in promoting more transparent and reliable AI systems.

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

## A  LIMITATIONS

While the relevancy metric provides a scalable and effective approach to quantifying the influence of training samples on test performance, it has certain limitations. First, the metric relies on the loss function's behavior, and its effectiveness is diminished when using discontinuous or highly non-smooth losses, such as 0-1 loss, which do not provide meaningful gradients. Second, relevancy captures correlations in learning dynamics but does not explicitly infer causal relationships between training and test samples, which may lead to misinterpretations in scenarios with complex indirect interactions. Lastly, the metric assumes the availability of sufficient training checkpoints to track loss dynamics, which could impose storage constraints in resource-limited environments. Addressing these challenges is an exciting avenue for future research to further enhance the robustness and applicability of the metric.

## B  ADDITONAL VISUALIZATIONS

### B.1  HIGH AND LOW RELEVANCY PAIRS

Figures 9 and 10 shows additional examples visualizing samples with high and low `Rel` values.

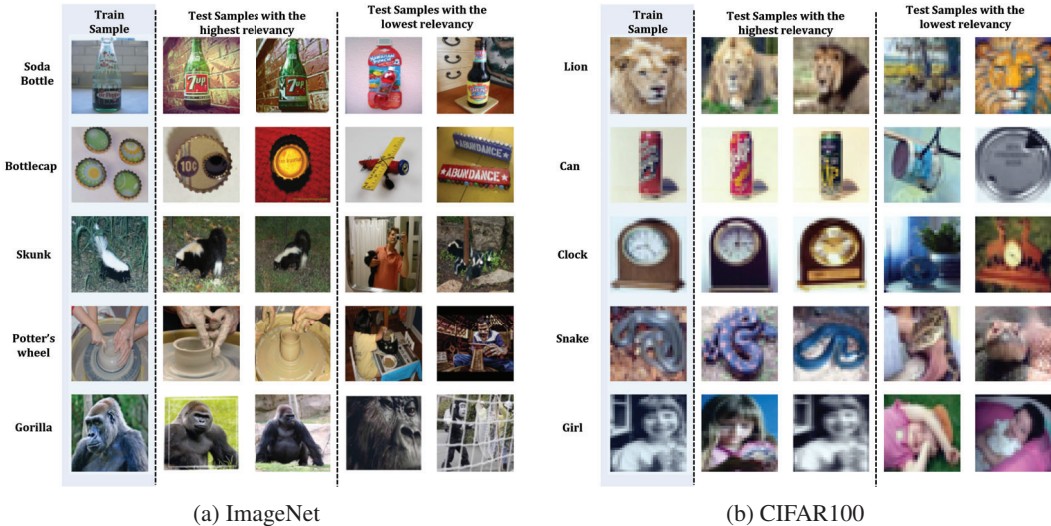

(a) ImageNet                                           (b) CIFAR100

Figure 9: Additional examples: Test samples with the highest and lowest `Rel` values for randomly selected train samples from randomly selected classes in popular computer vision datasets.

### B.2  PROTOTYPICAL AND ATYPICAL EXAMPLES

Figure 11 shows the most prototypical and most atypical samples of randomly selected classes. These are selected based on the `Atyp` scores.

### B.3  MEMORIZED SAMPLES DETECTED

Figure 12 shows examples of highly memorized samples (samples with the highest $Rel_{Mem}$ scores) from ImageNet, CIFAR-100, and Fashion-MNIST datasets. These samples tend to be atypical or have hidden features that make them hard to generalize, leading the model to memorize them instead. As discussed in Section 4.2, samples with high `mem` and `Curv` scores have high $Rel_{Mem}$ scores as well. Hence samples identified as memorized by those metrics, are identified by our metric as well.

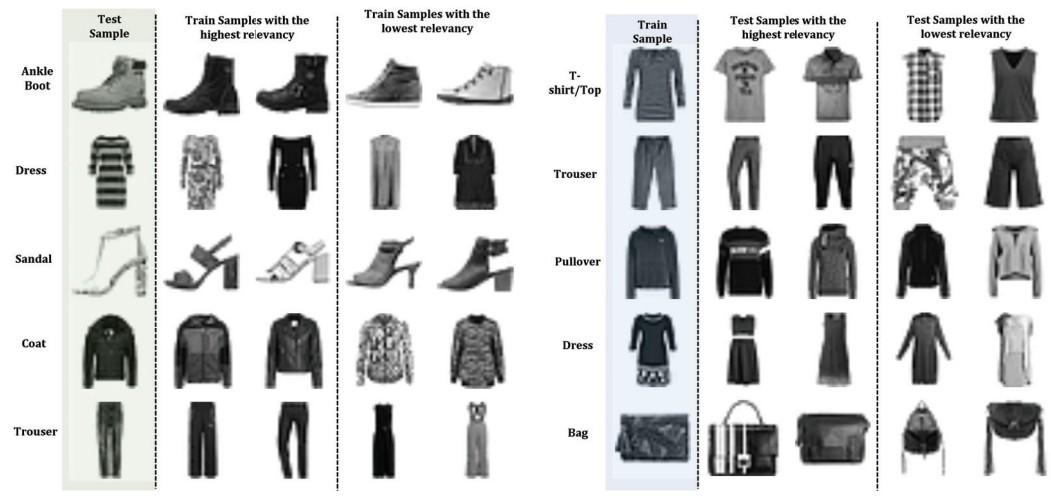

(a) Random Test Samples         (b) Random Train Samples

Figure 10: Additional examples: Samples with the highest and lowest `Rel` values for randomly selected train/test samples from randomly selected classes in the Fashion-MNIST dataset.

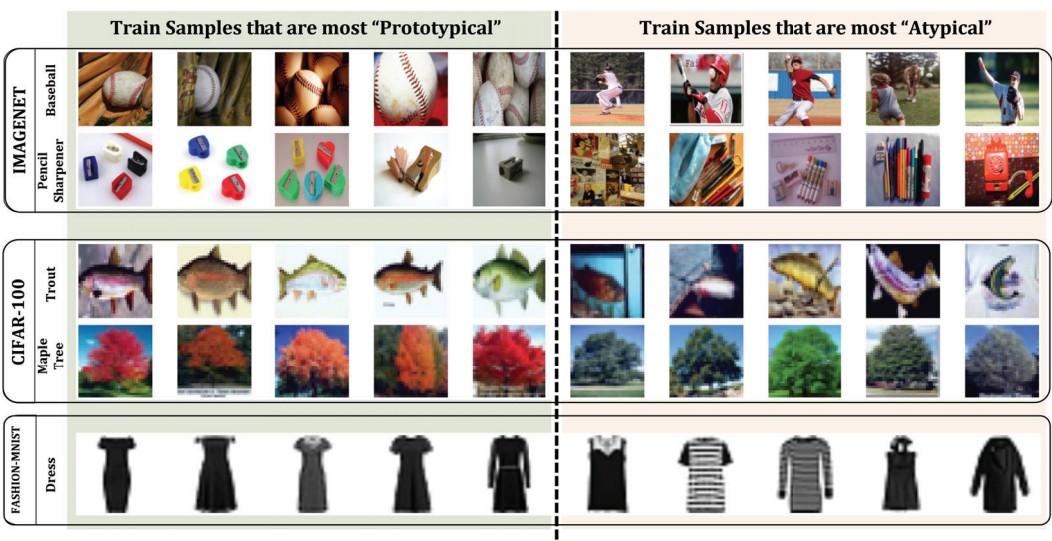

Figure 11: The 5 most "prototypical" (lowest `Atyp`) and 5 most "atypical" (highest `Atyp` samples of randomly selected classes in popular computer vision datasets

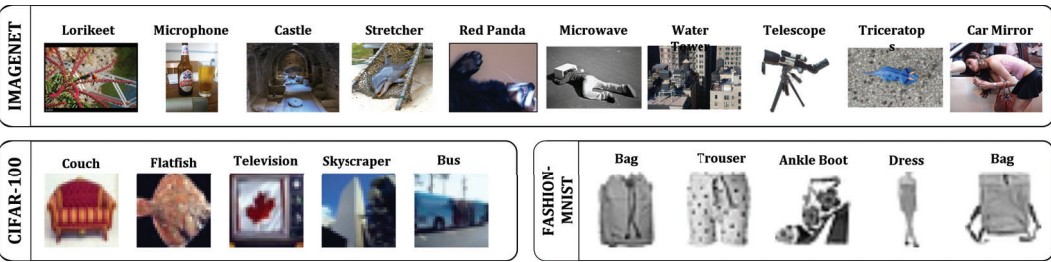

Figure 12: Examples of highly memorized samples (randomly selected) across classes from the training splits of popular computer vision datasets. The corresponding labels are indicated above each sample.

# C  DETAILED RESULTS FOR CORESET GENERATION EXPERIMENTS

All networks were trained using an SGD optimizer (Bottou, 2010) for 164 epochs with a learning rate of 0.1, scaled by 0.1 at epochs 81 and 121. Nesterov momentum (Sutskever et al., 2013), with momentum of 0.9, was turned on and a weight decay of 5e-4 was used. We used the following sequence of data augmentations for training: resize to $(32 \times 32)$, random crop with padding $= 4$, random horizontal flip, and normalization. 5 randomly seeded runs were run and the mean and variance of these runs are reported in Table 2.

No additional finetuning or regularizing was utilized for any of the coreset generation techniques (including ours). This was done to ensure complete fairness in comparison. All methods used the entire test set as the validation set to choose samples for the coreset.

**How our coresets were selected:** We first compute $\texttt{Rel}_{\texttt{Avg}}$ for all training samples using the test set available. Then samples per class were selected based on these scores.

Note that at higher coreset sizes, our method performs comparably to existing methods despite being computationally inexpensive.

Table 2: Performance summary of various coreset generation techniques on CIFAR-100 with ResNet-18. The mean values are shown in the first line and the standard deviations are shown in the second.

| Samples per Class | Random | Cal | GraphCut | Glister | GraNd | Forgetting | Herding | SloCurves | Ours $\texttt{Rel}_{\texttt{Avg}}$ |
|---|---|---|---|---|---|---|---|---|---|
| 1 | 3.48 | 5.24 | 4.80 | 3.43 | 2.49 | 3.52 | 3.33 | 2.65 | 5.28 |
|   | ±0.5 | ±0.4 | ±0.2 | ±0.3 | ±0.2 | ±0.2 | ±0.5 | ±0.3 | ±0.3 |
| 2 | 5.52 | 7.47 | 7.76 | 4.91 | 2.67 | 5.12 | 4.73 | 4.42 | 7.81 |
|   | ±0.5 | ±0.3 | ±0.4 | ±0.4 | ±0.3 | ±0.5 | ±0.5 | ±0.1 | ±0.3 |
| 3 | 7.12 | 9.12 | 8.96 | 5.49 | 3.01 | 6.02 | 4.97 | 5.51 | 9.14 |
|   | ±0.21 | ±0.3 | ±0.3 | ±0.6 | ±0.1 | ±0.2 | ±0.4 | ±0.2 | ±0.3 |
| 4 | 6.40 | 5.96 | 6.89 | 4.65 | 3.03 | 5.42 | 4.78 | 5.19 | 9.40 |
|   | ±0.7 | ±0.8 | ±0.8 | ±0.5 | ±0.5 | ±0.4 | ±0.7 | ±0.8 | ±0.3 |
| 5 | 8.19 | 10.19 | 10.86 | 6.04 | 3.68 | 7.60 | 6.28 | 7.68 | 12.10 |
|   | ±0.3 | ±0.4 | ±0.5 | ±0.4 | ±0.2 | ±0.4 | ±0.3 | ±0.2 | ±0.4 |
| 10 | 12.45 | 13.72 | 13.95 | 8.11 | 4.67 | 9.90 | 8.98 | 12.66 | 17.66 |
|   | ±0.5 | ±0.3 | ±0.6 | ±0.4 | ±0.1 | ±0.2 | ±0.5 | ±0.3 | ±0.7 |
| 15 | 16.14 | 15.45 | 14.98 | 9.52 | 5.71 | 12.30 | 9.42 | 16.84 | 22.10 |
|   | ±0.9 | ±1.0 | ±0.6 | ±0.3 | ±0.3 | ±0.3 | ±0.5 | ±0.9 | ±0.6 |
| 20 | 19.10 | 17.05 | 16.21 | 10.57 | 6.33 | 13.83 | 11.64 | 20.68 | 24.75 |
|   | ±1.1 | ±1.0 | ±0.7 | ±0.6 | ±0.5 | ±0.9 | ±1.2 | ±0.6 | ±0.5 |
| 25 | 22.60 | 17.97 | 17.33 | 11.47 | 7.11 | 14.38 | 12.00 | 26.01 | 28.10 |
|   | ±1.0 | ±0.5 | ±1.6 | ±0.7 | ±0.5 | ±0.8 | ±1.4 | ±0.6 | ±0.5 |
| 30 | 25.77 | 19.93 | 19.31 | 12.99 | 8.01 | 15.46 | 15.10 | 26.91 | 30.81 |
|   | ±1.1 | ±0.5 | ±0.7 | ±0.7 | ±0.2 | ±0.6 | ±0.9 | ±1.4 | ±0.6 |
| 35 | 27.82 | 21.37 | 21.35 | 13.83 | 8.40 | 15.65 | 14.93 | 33.07 | 34.03 |
|   | ±1.2 | ±0.8 | ±0.7 | ±0.8 | ±0.5 | ±0.7 | ±0.6 | ±0.9 | ±1.2 |
| 40 | 31.57 | 22.32 | 22.63 | 14.77 | 8.47 | 17.25 | 16.45 | 34.55 | 35.69 |
|   | ±1.3 | ±1.1 | ±1.5 | ±0.8 | ±0.5 | ±1.2 | ±0.4 | ±0.9 | ±1.3 |
| 45 | 33.93 | 23.64 | 26.22 | 14.54 | 9.66 | 19.45 | 18.18 | 38.23 | 38.09 |
|   | ±0.9 | ±1.5 | ±0.8 | ±1.3 | ±0.4 | ±0.9 | ±1.0 | ±1.4 | ±0.9 |
| 50 | 34.09 | 23.19 | 22.41 | 15.22 | 10.45 | 18.45 | 16.80 | 37.57 | 37.63 |
|   | ±0.8 | ±1.0 | ±2.0 | ±0.3 | ±0.5 | ±0.6 | ±0.9 | ±1.5 | ±0.9 |

# D   SYNTHETIC NOISE EXPERIMENTS

To evaluate the robustness of the relevancy metric ($Rel_{Mem}$) in detecting noisy or irrelevant samples, we conducted an experiment by introducing $10\%$ synthetically corrupted noisy samples into the training dataset. These noisy samples were designed to simulate mislabeled or irrelevant data, testing the metric's ability to distinguish them from regular samples.

**Setup**   We randomly corrupted $10\%$ of the training samples by assigning them incorrect labels. The training process was carried out as usual, and the ($Rel_{Mem}$) scores were computed for all training samples based on the model's learning dynamics.

**Results**   Figure 13 shows the histogram of $Rel_{Mem}$ scores for the dataset. The synthetically corrupted noisy samples are marked as red crosses. As observed, these noisy samples exhibit significantly higher $Rel_{Mem}$ scores compared to the majority of regular samples. This strong separation demonstrates the metric's ability to identify noisy samples effectively.

**Discussion**   These results confirm that the relevancy metric reliably assigns higher scores to noisy or mislabeled samples, reflecting their memorization behavior during training. This highlights the metric's robustness in detecting irrelevant samples that may hinder model generalization.   This

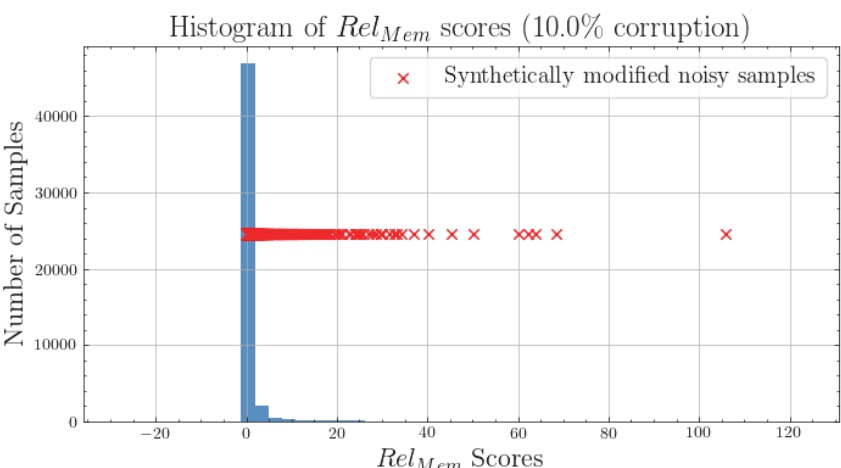

Figure 13: Histogram of $Rel_{Mem}$ scores with $10\%$ synthetically corrupted noisy samples. The noisy samples (red crosses) exhibit significantly higher scores compared to the majority of regular samples.

experiment demonstrates the utility of the relevancy metric in identifying noisy samples and supports its application to diverse datasets.

# E   ABLATION STUDY - SKIPPING EPOCHS DURING RELEVANCY CALCULATION

The core principle of *relevancy* lies in observing the dynamics of the loss of samples during training. Hence it is important to utilize the loss values at each epoch for a more nuanced `Rel` metric. It is however possible to downsample the number of loss values (by only considering some of the epochs) to still get a meaningful *relevancy* score. Figure 14 illustrates the effect of downsampling, by plotting the histograms of the `Rel` scores of all classes. As the downsampling factor increases, the `Rel` values move towards 1.0. This is because there is not enough information to differentiate the learning dynamics of various samples, and all samples will appear to be learned (loss values decrease) at the same time.

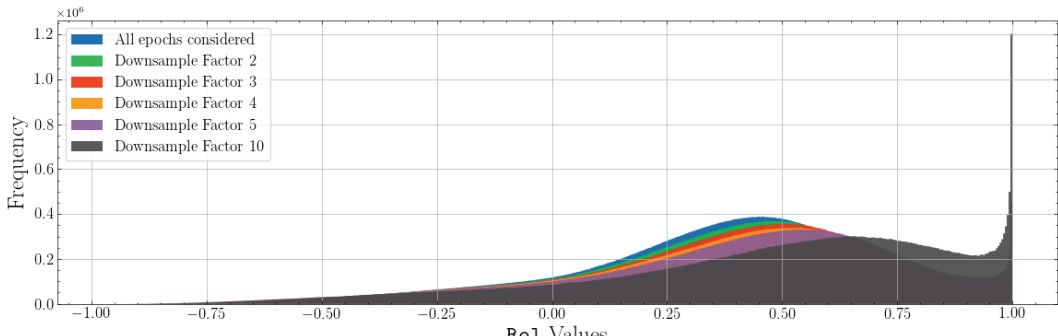

Figure 14: Histogram of `Rel` scores (of all classes) of ImageNet illustrating the effect of downsampling loss values (by skipping epochs).

## F DETAILED COMPARISON WITH EXISTING METRICS

**Recap on Notation:** We denote the number of training samples as $m$ ($S \sim \mathbf{Z}^m$) and the number of test samples as $m'$. The model is trained for $T$ epochs, and for certain methods, there are hyperparameters (denoted as $r$) that further influence the computational cost. The computational complexity of each method is expressed in terms of floating point operations per second (FLOPs), denoted as $f$, which represent the cost of a single forward pass for a model with $p$ parameters. The cost of a backward pass is approximately $2f$ FLOPs.

`Infl` **&** `mem` **Scores:** These scores were proposed by Feldman & Zhang (2020) and Feldman (2020) respectively.

The influence score `Infl` of a training sample $\vec{z}_i$ on a test sample $\vec{z}_j$ is given by:

$$\mathtt{infl}(\mathcal{A}, S, i, j) = \Pr_{\mathcal{A}(S,T)}[h_S^{\phi,T}(\vec{x}_j) = y_j] - \Pr_{\mathcal{A}(S^{\backslash i},T)}[h_{S^{\backslash i}}^{\phi,T}(\vec{x}_j) = y_j]$$

Here, $\Pr_{\mathcal{A}(S,T)}$ represents the confidence in the prediction for $\vec{z}_j$, and $S^{\backslash i}$ is the training set with sample $\vec{z}_i$ removed. This expression measures the change in the prediction probability for the test sample $\vec{z}_j$ when the training sample $\vec{z}_i$ is included in the training set compared to when it is excluded. The underlying intuition is that if sample $\vec{z}_i$ significantly influences the prediction for $\vec{z}_j$, its exclusion will lead to a notable reduction in prediction accuracy for $\vec{z}_j$, resulting in a higher influence score.

Similarly, `mem` scores are defined as

$$\mathtt{mem}(\mathcal{A}, S, i) = \Pr_{\mathcal{A}(S,T)}[h_S^{\phi,T}(\vec{x}_i) = y_i] - \Pr_{\mathcal{A}(S^{\backslash i},T)}[h_{S^{\backslash i}}^{\phi,T}(\vec{x}_j) = y_i]$$

Both these metrics ideally require retraining $m+1$ models to calculate the memorization scores of all $m$ samples of the training dataset $S$. To reduce the retraining costs, the authors of these metrics suggest training a smaller number ($r$) of models with a fraction ($s$) of the training dataset. Each of these subsets is selected randomly. They then estimate $\Pr_{\mathcal{A}(S,T)}[h_S^{\phi,T}(\vec{x}_i) = y_i]$ to be the average prediction probability of $\vec{z}_i$ of all models that were trained with a subset $S'$ that included the sample $i$ ($i \in S'$), and $\Pr_{\mathcal{A}(S^{\backslash i},T)}[h_{S^{\backslash i}}^{\phi,T}(\vec{x}_i) = y_i]$ to be the average prediction probability of $\vec{z}_i$ of all models that were trained with a subset $S''$ that excluded the sample $i$ ($i \notin S''$).

Hence the total computational overhead would be to train $r$ models for $T$ epochs (run forward and backward passes for each sample $T \cdot r$ times ), and for each model, we need to calculate the prediction probability (or run a forward pass) for $s \cdot m$ train samples and $m'$ test samples. The total computational overhead for `Infl` scores are $(T \cdot s \cdot m \cdot 3f + m' \cdot f) \cdot r$.

Similarly, the computational overhead for `mem` scores are $T \cdot s \cdot m \cdot 3f \cdot r$, as no additional forward passes are required for the test samples.

`TracIn` **Score:**  This metric was proposed by Pruthi et al. (2020). It computes the influence as:

$$\texttt{TracIn}(\mathcal{A}, S, i, j) = \sum_{t=0}^{T} \nabla \ell(h_S^{\phi,t}, \vec{z}_i) \cdot \nabla \ell(h_S^{\phi,t}, \vec{z}_j)$$

This metric requires computing the gradient (one forward and one backward pass) for each sample in the train set of size $m$, and test set of size $m'$ for each epoch ($T$ times). Hence the computational overhead is $T \cdot (m' + m) \cdot 3f$.

`Curv` **Score:**  This metric was proposed by Garg et al. (2024). It computes the input curvature, a proxy for memorization as :

$$\texttt{Curv}(\mathcal{A}, S, i) = \frac{1}{rT} \sum_{t=0}^{T} \sum_{e=1}^{r} \left\| \frac{\partial \left( \ell(h_S^{\phi,t}, \vec{z}_i + hv) - \ell(h_S^{\phi,t}, \vec{z}_i) \right)}{\partial \vec{z}_i} \right\|_2^2$$

Here, the hyperparameters $r$ represent the number of "repeats" done to get an empirical expectation of the randomly generated $hv$ which represents the Rademacher random variables used in Hutchinson's trace estimator (Hutchinson, 1989).

For each epoch $T$, each training sample ($m$), and each repeat $r$, 2 forward passes and 1 backward pass are required. Assuming that $\ell(h_S^{\phi,t}, \vec{z}_i))$ can be estimated only once, this requires a total of $r + 1$ forward and $r + 1$ backward passes for each sample per epoch. Hence the total computational overhead is $T \cdot (1 + r) \cdot m \cdot 3f$

