# OpenReview forum: "The Relevancy Metric: Understanding the Impact of Training Data"
_ICLR.cc/2025/Conference — Submitted to ICLR 2025_

### Official Review · Reviewer_WqAr · 2024-11-02

**Soundness:** 3
**Presentation:** 3
**Contribution:** 2
**Rating:** 3
**Confidence:** 4

**Summary:**

The authors introduce a novel metric termed 'relevancy,' which aims to measure the influence of individual training samples on predictions.
Essentially, relevancy is derived by the similarity between a training instance and a testing point on the training loss during the training process.
The metric is computationally efficient, provided access to the model's checkpoints at each training epoch is available.
The authors leverage this relevancy metric to construct a coreset and demonstrate its utility in identifying mislabeled samples within datasets.

While the proposed approach is intriguing, there are aspects that could benefit from further elucidation.
My major concern pertain to the metric's underlying insights, particularly in scenarios where relevancy might fail to show actual influence.
Additionally, the paper's experimental section does not sufficiently address these concerns, preventing me from endorsing its acceptance at this stage.

**Strengths:**

Contributions:
1. The authors introduce a relevancy metric grounded in the similarity between training samples and testing points, offering a means to assess the contribution of individual samples to test predictions.
2. The metric is computationally efficient, assuming access to model checkpoints throughout the training process.
3. Utilizing the relevancy metric, the authors construct a coreset and achieve state-of-the-art results on CIFAR-100.
4. They also demonstrate the metric's potential in detecting mislabeled samples within datasets.
5. The paper is well-structured and clearly written.
6. The comparison with other metrics is useful.

**Weaknesses:**

Flaws:

1. [Major Concern] The metric's reliability is questionable, as it relies solely on loss information.
In extreme cases, such as when using 0-1 loss, the metric could be highly unstable.
There may be instances where high relevancy does not equate to significant contribution to the testing point, and vice versa, yet the paper lacks discussion on these scenarios.
The authors should provide insights into the conditions under which the metric is effective and when it is not. Without this, it is challenging to advocate for its use on new datasets.

2. Building on the first point, if the authors do not clarify the metric's applicability, the current experiments are insufficient. Additional experiments showcasing the metric's utility across various models and datasets are needed to establish its broad usefulness.

3. Would the selected coreset overfit to test points? Did the authors validate the coreset against a separate test set or use validation samples to derive the coreset?

4. What is the implication of negative relevancy? Does it suggest that training samples negatively impact the testing point?


Small concerns:
1. Figure 2 is unclear and difficult to read.

**Questions:**

See above.

---

> ### Author Response · Authors · 2024-11-25
>
> We thank the reviewer for their feedback and aim to address the concerns raised below.
>
> 1. We appreciate the reviewer’s thoughtful comment regarding the reliability of the relevancy metric, especially in extreme cases and its reliance on loss information. While the relevancy metric does rely on loss dynamics, its correlation-based formulation mitigates the instability caused by extreme loss values, as it focuses on the relative changes in loss across training rather than absolute values. However, we agree that specific cases, such as using discontinuous loss functions like 0-1 loss, could lead to instability or unintended behavior. In such cases, relevancy’s effectiveness may be diminished, as the metric depends on continuous and meaningful loss gradients to capture train-test relationships.
>
>    To provide more clarity, the metric is most effective when the loss function is smooth and reflects generalization behaviors, such as cross-entropy or mean squared error, which are commonly used in modern deep learning tasks. In contrast, tasks with discontinuous loss functions or highly adversarial data distributions might require additional adaptations or complementary methods to ensure reliability.
>
>    We have included a section in the appendix (Appendix A, highlighted in red) to outline these limitations and the conditions where relevancy performs best. Future work could explore extending the metric to account for these edge cases, enhancing its robustness for diverse datasets and loss functions.
>
> 2. We tested and showcased the utility of the relevancy metric across model architectures (in Section 62.) and across datasets (in Section 6.1).
>
> 3. We used a separate validation set to select the coreset as explained in Section 5.
>
>
> 4. Yes, a training sample with a negative relevancy score for a testing point suggests that the training sample negatively impacts the test sample's prediction. This occurs when the features learned from the training sample during training conflict with or diverge from the features of the testing sample during inference.
>
>    For example, a training sample of a white tiger might negatively impact a test sample of a regular (orange) tiger. During training, the white tiger sample could influence the model by associating the "tiger" class with white and black coloration (simplified for explanation). During inference, this learned association may conflict with the features of the regular tiger test sample, which lacks white and black coloration, reducing the model's confidence or accuracy for the test sample.

---

> > ### Comment · Reviewer_WqAr · 2024-11-27
> >
> > I thank the authors for the responses.
> > After reading the appendix provided by the authors, regarding the reliability part, I am still confused on when and how this new metric works. It might need more analysis and provide more details, either on the experimental or theoretical parts.
> > I also read the other reviews carefully, and feel that this paper is not ready to be published in ICLR at the time being.
> > Therefore I remain my score unchanged.
> > However, I do think that this paper focuses on an important topic, and I encourage the authors to polish this manuscript and submit it to the next conference.

---

### Official Review · Reviewer_fUUz · 2024-11-03

**Soundness:** 1
**Presentation:** 4
**Contribution:** 1
**Rating:** 3
**Confidence:** 5

**Summary:**

This paper proposes relevancy, which quantifies each training data's attribution for model's inference.
Relevancy tracks how correlated the train data loss is to that of the target data during training, and evaluate the data as relevant if positively correlated, and vice versa.
Such suggested metrics are computationally efficient, and adopted to various applications such as dataset compression and detecting mislabeled data.

**Strengths:**

This paper is well-written and addresses the critical research question of attributing training data to model performance, especially as the significance of data continues to grow in the era of large-scale models.

**Weaknesses:**

**1. Missing important baselines**: This paper lacks important previous studies that tackle the same goal, i.e., training data attributions (TDA) [1,2]. These work evaluates TDA via training data's counterfactual on model's output on target data. Also, this paper lacks the evaluation on existing metric of TDA, called linear datamodeling score (LDS), as proposed in Park et al.[2]

**2. Wrong definition**: The proposed relevancy, which measures correlation between train and target data's losses, does NOT mean the influences of train data onto the target data. In other words, train data is simply correlated, but not causal, to target data's reduced loss. For instance, there is the true causal train data (i.e., *tr1*) that helps minimize both (another) train data (i.e., *tr2*) and target data (i.e., *te*), while training on *tr2* does not induces lower loss on *te*. Still, the proposed relevancy will identify *tr1* as strongly influencing train data for *te*, which is not true. Therefore, I'm not convinced that the proposed Relevancy is capable of measuring true influence of training data to the target data.

**3. Incomplete analysis**: While authors claim that Fig.4-(b) reveals "meaningful relationships" across all test samples, it is unclear what meaningful relationships regarding true influences between data are represented in the figure.

**4. Potential computational complexity**: Since Relevancy relies on loss, it might be highly stochastic across different training setup, e.g., data orders, architectures, learning rates. How invariant or reliable the Relevancy is given such stochasticity? Is just a single run sufficient to obtain reliable evaluation, or it requires multiple iterative runs for reliable results that would result in additional computational complexity, unlike stated in Table 1.?


[1] Ilyas et al., Datamodels: Predicting Predictions from Training Data, ICML 2022 \
[2] Park et al., TRAK: Attributing Model Behavior at Scale, ICML 2023

**Questions:**

See weaknesses.

---

> ### Author Response · Authors · 2024-11-25
>
> We thank the reviewer for their feedback and aim to address the concerns raised below.
>
> 1. We thank the reviewer for pointing out relevant prior work on training data attributions (TDA), particularly [1] and [2]. These studies, including the datamodeling approach by Ilyas et al. [1] and the linear datamodeling score (LDS) proposed in Park et al. [2], are indeed closely related to our goal of understanding the influence of training data on model behavior. We acknowledge that these works were not explicitly discussed in our paper and appreciate the opportunity to address this oversight.
>
>    Both datamodels and LDS use counterfactual approaches to evaluate training data influence, offering valuable insights into training-test relationships. However, these methods often involve additional computational complexity and require specific assumptions (e.g., model linearity or surrogate models). In contrast, our relevancy metric provides a lightweight and scalable alternative by directly leveraging the learning dynamics during training, making it broadly applicable without requiring explicit counterfactual constructions.
>
>    We have incorporated such a discussion on datamodels and LDS in the Section 2 (highlighted in red in the revised manuscript) to highlight these connections and position relevancy as a complementary approach to existing TDA methods.
>
> 2. We appreciate the reviewer’s observation regarding the distinction between correlation and causation in relevancy. To clarify, relevancy is designed to measure the relationship between training samples and test examples, capturing correlations in learning dynamics rather than strict causal inference. While this does not establish direct causality, it provides actionable insights into how training samples influence generalization.
>
>    In the scenario with tr1 and tr2, relevancy would capture the relationship between tr1 and te indirectly via tr2, reflecting their shared optimization dynamics. This aligns with the metric’s goal of identifying samples that play a significant role in shaping the model’s predictions, even indirectly.
>
>    Relevancy is a practical and computationally efficient tool for understanding train-test relationships and complements more complex methods focused on causality. We acknowledge the distinction and see future work as an opportunity to disentangle these effects further.
>
> 3. Influence scores measure the change in prediction probability when a sample is included in the training dataset to when it is not. Since many samples assist in making the model successfully predict on an inference sample, exclusion of a sample may not have a huge effect. This bimodal distribution of influence scores where most values are 0 or close to 1 has been extensively studied in literature. It is hard to utilize this information to generate coresets from this information as most values are 0. However, our metric assigns a non-zero value to most train-test pairs and is a more continuous distribution. Hence, we can leverage this more successfully in applications such as generating coresets. This is why we say the relevancy metric provides more of an insight (meaningful relationship) than influence scores.
>
> 4. The reviewer is right that due to the stochastic nature of training, each run may provide slightly different loss trajectories. We take this into account and show a detailed study on how the randomness of the training (seed) or the model architectures do not affect the relevancy metric in Section 6.2.

---

> > ### Comment · Reviewer_fUUz · 2024-12-03
> >
> > 1. Thank the authors for adding detailed discussions on the previous studies on the updated script. However, these methods are one of the core baselines that are missed in this paper, even if they require the computational cost. My concern on such missing baselines are not addressed.
> > 2. I do not agree with the authors' claim that "while this does not establish direct causality, it provides actionable insights into.." How could one ensure one data's impact on target data's without establishing its causality? The proposed metrics that do not clearly differentiate the causality and correlation lead to misinterpretation of the training data's effect on target data (task).
> > 3. Again, using the previous well-established metric for measuring the attribution of the training data --- LDS --- should be included for fair comparison.
> >
> > Overall, I believe this paper needs (1) clarification on the definition on how the impact of train data, and (2) comparisons with existing baselines of training data attributions. Therefore, I keep my rating as initial review.

---

### Official Review · Reviewer_n349 · 2024-11-05

**Soundness:** 2
**Presentation:** 3
**Contribution:** 2
**Rating:** 3
**Confidence:** 4

**Summary:**

This paper introduces a novel metric called Relevancy to quantify the impact of individual training samples on model predictions. The metric aims to efficiently capture the relationship between different samples by analyzing the loss trajectories over epochs, allowing insights into data memorization, generalization, and the identification of mislabeled samples. The authors demonstrate that the proposed metric is computationally efficient—up to 100,000× faster than traditional metrics like influence functions and input curvature—and effective in generating coresets that improve classification accuracy on CIFAR-100.  This paper also demonstrates mislabeled samples detection use case for relevancy.

**Strengths:**

1.	This paper is well-organized and easy to read. The authors first introduce the proposed metric and then provide a comprehensive comparison with other alternative metrics in terms of score results and computation cost. Subsequently, several use cases, such as coreset generation and mislabeled sample identification, and consistency investigation are provided to show the versatility of this metric.
2.	The proposed metric relevancy is simple, efficient, and computation-friendly. The Relevance metric is a simple approach that captures the impact of individual training samples through correlation in loss trajectories. The computational efficiency is well-documented, showing up to 100,000× improvement over other metrics.
3.	Relevancy’s applications in coreset selection and mislabeled sample identification highlight its broad relevance and potential for practical use in dataset management, model interpretability, and error reduction.

**Weaknesses:**

1.	Can the authors provide more details about the impact of individual training samples on inference prediction? What kind of impact does this paper aim to quantify? Any formal definition on this impact? From my understanding, the impact this paper aims to quantify is different from that using the influence function. The train-test relationship depends on the sample pair, the well-trained model, and the target loss while relevancy depends on the learning dynamics.
2.	Although the relevancy definition is simple, I have concerns about the definitions using correlation. (1) why is the learning dynamic involved in impact quantification? Suppose there are two identical well-trained models but with different training trajectories, why are the train-test relationship scores different from these two models? (2) For large training epochs T, if we continue to train the model with one epoch in a different task, the relevancy score should not change significantly since the correlation coefficient is dominated by the loss curve in the first large T epochs. However, the model at T+1 epoch could be significantly different from the model at T epoch and thus with different impact for train-test sample pairs. (3) Since the relevancy metric only considers loss values, it may not be dominated by some huge loss value. The robustness may be a potential issue. (4) The biggest concern is about the order preservation among loss transformation. Suppose two train-test pairs with the same relevancy (e.g., loss value A1=[2.0, 1.0, 0.5], B1=[4.0, 2.0, 1.0], and A2 = [3, 2, 1] B2=[6, 4, 2]), if we use different loss mapping (e.g., CE and perplexity), the relevancy order could be changed. Overall, it would be better to start from some principles to design such metrics and then justify the proposed metric by satisfying these principles.
3.	The limitation discussion of this paper is missing. The relevancy score computation requires the storage of all model checkpoints for all epochs, which narrows the applicability of this metric.
4.	Experiments can be more convincing. (a) There are no experiments to evaluate the relevancy score quality. It would be better to design some synthetic experiments. For example, the impact of a training example on a pure noise image should be very small. What’s the relevancy score and other metrics score in this case? (b) Why can the proposed metric achieve the best performance than other coreset generation methods? Any possible reason/rationale for this? (c) For noisy sample detection, I am curious whether this method can still perform well in high noise ratio regime. Considering the extreme case with almost 100% noise ratio for some class, the average relevancy score may be dominated by noisy samples and cannot reflect the real impact of the sample. (d) It would be better to compare the consistency over different model backbones for different metrics. Does relevancy show higher consistency than influence function?

While my current score is low, I am open to revising it if the authors can address my concerns.

**Questions:**

Please see the weakness part.

---

> ### Author Response · Authors · 2024-11-25
>
> We thank the reviewer for their detailed feedback and constructive criticism. Below, we address the concerns and clarify the proposed relevancy metric.
>
> The relevancy metric evaluates the influence of a training sample on a sample of interest (e.g., test or other training samples) for a specific task. It quantifies this impact dynamically by correlating the loss changes of the training sample with those of the sample of interest across training iterations. Unlike influence functions, which measure the static effects of perturbing or removing samples, the relevancy metric captures the evolving contribution of training samples to both memorization and generalization during learning. This dynamic perspective provides actionable insights into train-sample interactions, enabling tasks such as coreset selection, noisy data detection, and understanding model behavior.
>
> **1. Impact of Individual Training Samples and Formal Definition:** The relevancy metric formally quantifies the dynamic impact of training samples by analyzing the correlation between their loss changes and those of the sample of interest across training iterations. This approach reveals how training samples influence the model during training, providing richer insights than static methods like influence functions. For example, samples with higher relevancy positively contribute to generalization, while low-relevancy samples indicate noise or atypical patterns. This is further demonstrated in tasks such as coreset selection and mislabeled sample detection.
>
> **2. Concerns Regarding Correlation-Based Definitions**
>
>    1. Observing learning dynamics is crucial for understanding sample-level interactions, as supported by prior works [1-4]. Relevancy scores remain consistent across models with identical tasks, irrespective of differences in learning hyperparameters, architectures, or initialization seeds. We demonstrate this in Section 6.2.
>
>    2. The metric is task-specific and reflects loss dynamics within the training trajectory for a given task. Extending the model to a new task (e.g., after epoch T+1) requires recomputation, similar to other metrics in this domain.
>
>    3. While relevancy considers changes in loss values during training, it inherently captures the relationship between losses of training and test samples rather than being directly influenced by the magnitude of individual losses. This ensures that outliers with disproportionately large loss values do not dominate the metric. Additionally, the relevancy's averaging and correlation-based nature mitigates such outliers' impact.
>
>    4. Since relevancy is tied to task-specific loss functions, its computation is naturally dependent on the chosen loss. Differences in loss transformations (e.g., cross-entropy vs. perplexity) could affect order preservation. Future work could explore principled loss normalization techniques to address this.
>
> **3. Limitations of the Metric:** We acknowledge the reviewer’s concern regarding the memory-intensive nature of storing multiple checkpoints. We will include this limitation in the paper. However, relevancy can be computed on smaller models and reused for tasks like coreset generation in larger models, making it computationally efficient compared to retraining-based metrics. As shown in Table 1, relevancy significantly reduces computational overhead compared to influence functions.

---

> ### Author Response · Authors · 2024-11-25
> **(Continued)**
>
> **4. Experimental Validation**
>
>    - a) The relevancy metric theoretically assigns near-zero scores to pure noise inputs due to their lack of correlation with meaningful training data. Our existing results demonstrate robustness in identifying mislabeled and atypical samples, supporting its reliability in real-world settings. Synthetic experiments to further validate this behavior are explained in part c of this response.
>
>    - b) Relevancy outperforms other coreset selection methods by directly quantifying the evolving impact of training samples on test generalization through loss dynamics. In contrast, other methods often rely on static heuristics, which may not effectively capture the dynamic train-test relationship.
>
>    - c) High-Noise Regime: We appreciate the reviewer’s suggestion to evaluate the relevancy score with synthetic experiments. We introduced synthetically modified noisy samples (10% corruption) into the training data and analyzed their $Rel_{Mem}$ scores to address this. We have included the full experimental details and analysis in appendix D (highlighted in red) to demonstrate the robustness and reliability of the relevancy metric. The histogram (Figure 13 in the revised manuscript) shows these noisy samples exhibit significantly higher $Rel_{Mem}$ scores (marked as red crosses), making them distinguishable from regular samples. This validates the metric’s ability to identify noisy and atypical samples effectively.
> Furthermore, the design of the relevancy metric ensures that it captures meaningful correlations between training and test samples, with noise inputs expected to yield higher scores due to their inherent tendency to be memorized during training. This behavior is consistent with prior results in our paper, where the metric successfully identifies mislabeled and memorized samples in real-world datasets. While we agree that further exploration of edge cases could strengthen the evaluation, we believe this experiment, combined with existing results, sufficiently demonstrates the robustness and practical utility of the metric.
>
>    - d) Relevancy has shown consistency across different model backbones, as demonstrated in Section 6.2. The metric is task and loss-dependent and differs for each task. While influence functions could provide a useful comparison, their computational expense (due to multiple retrainings) limits their practicality in such evaluations.
>
> **References:**
>
> *[1] Mariya Toneva, Alessandro Sordoni, Remi Tachet des Combes, Adam Trischler, Yoshua Bengio, and Geoffrey J Gordon. An empirical study of example forgetting during deep neural network learning. In International Conference on Learning Representations, 2018.*
>
> *[2] Karttikeya Mangalam and Vinay Uday Prabhu. Do deep neural networks learn shallow learnable examples first? In ICML 2019 Workshop on Identifying and  Understanding Deep Learning Phenomena, 2019*
>
> *[3] Ziheng Jiang, Chiyuan Zhang, Kunal Talwar, and Michael C Mozer. Characterizing structural regularities of labeled data in overparameterized models. In Marina Meila and Tong Zhang (eds.), Proceedings of the 38th International Conference on Machine Learning, volume 139 of Proceedings of Machine Learning Research, pp. 5034–5044. PMLR, 18–24 Jul 2021.*
>
> *[4] Isha Garg, Deepak Ravikumar, and Kaushik Roy. Memorization through the lens of curvature of loss function around samples. In Proceedings of the 41st International Conference on Machine Learning, volume 235 of Proceedings of Machine Learning Research, pp. 15083–15101. PMLR, 21–27 Jul 2024*

---

> > ### Comment · Reviewer_n349 · 2024-11-26
> > **Thanks for the response**
> >
> > Thank you to the authors for the detailed response. After reviewing the rebuttal, I appreciate the effort in addressing W3, which I believe is now well-handled. However, my primary concerns remain unresolved, and I will maintain my score while increasing my confidence to 4. My concerns are outlined as follows:
> >
> > - Clarity on the Impact Quantified by the Relevancy Metric. It remains unclear what specific type of impact the paper aims to quantify using the relevancy metric. For instance, with influence functions, the impact is explicitly defined as the change in target loss when a training sample is excluded. The rationale behind the relevancy metric's dependency on learning dynamics is also not well-justified.
> > - Concerns Regarding the Relevancy Definition. The rationale for incorporating dynamics into the relevancy metric remains unconvincing. The metric appears to lack robustness at large epochs, where it could be dominated by large loss values, potentially leading to the loss of rank order consistency.
> > - Lack of Experimental Evaluation of the Relevancy Metric. There are no experiments provided to evaluate the quality of the relevancy score, which is critical given the absence of ground truth or a formal definition of the impact the paper seeks to quantify. Furthermore, a comparison with influence functions in terms of consistency across models and tasks is missing, which would strengthen the paper's claims.

---

> > > ### Author Response · Authors · 2024-11-26
> > >
> > > We appreciate the reviewer’s feedback and their recognition of our efforts to address W3. Below, we respond to the key concerns:
> > >
> > > **1. Clarity on the Impact Quantified by the Relevancy Metric:** The relevancy metric captures the relationship between training and test samples by quantifying how their loss dynamics correlate during training. Unlike influence functions, which measure counterfactual removal, relevancy provides a lightweight alternative that reflects contributions in real time without requiring costly retraining. This paper establishes a starting point for leveraging learning dynamics as a practical and scalable measure of training-test interactions, distinct from static influence-based methods.
> > >
> > > **2. Concerns Regarding Robustness:** While the relevancy metric depends on loss values, it operates on relative correlations rather than absolute magnitudes, mitigating concerns about rank-order consistency at later epochs.
> > >
> > > **3. Experimental Evaluation and Comparison with Influence Scores:** Direct comparisons with influence functions across tasks are infeasible due to resource constraints, as influence scores cannot be computed for large datasets within our available resources. For tasks such as ImageNet, we rely on pre-computed scores provided by the authors of prior works. Using these, we have compared relevancy to influence scores on ImageNet (in Section 4.1) and demonstrated that relevancy remains consistent across models (in Section 6.2). These results support the robustness of our metric.

---

### Official Review · Reviewer_jfAD · 2024-11-08

**Soundness:** 3
**Presentation:** 4
**Contribution:** 3
**Rating:** 6
**Confidence:** 3

**Summary:**

This paper introduces "Relevancy," a novel metric to evaluate the influence of individual training samples on model predictions. Relevancy measures the correlation between the loss trajectories of training and test samples, aiming to understand how specific training samples impact generalization to unseen data.

**Strengths:**

1. The paper is well written and organized.
2. The proposed metric is both simple and innovative, providing an effective way to measure the influence of training samples on the model’s generalization capabilities. This simplicity does not compromise its effectiveness, making it broadly applicable across tasks that require interpretability and efficient data utilization.
3. The paper demonstrates the versatility of the proposed metric through its applications in coreset generation, memorization analysis, and mislabeled sample detection. The experimental results are thorough, covering multiple datasets and convincingly supporting the metric's utility.

Overall, this paper makes a meaningful contribution to deep learning research by proposing a practical and impactful metric.

**Weaknesses:**

1. The metric relies on loss trajectory data from training checkpoints, which could limit its use in models or systems where saving and processing frequent checkpoints is impractical or unavailable. Could you address potential methods to mitigate this limitation, such as using fewer checkpoints or approximations? How does the length of the trajectory T impact the metric?

2. Generalization Across Tasks: While the paper demonstrates the effectiveness of Relevancy, its performance on tasks beyond classification (e.g., regression or sequential tasks) is not explored. Testing the metric's adaptability to a wider range of applications would enhance its impact.

**Questions:**

see weakness.

---

> ### Author Response · Authors · 2024-11-25
>
> We thank the reviewer for their thoughtful feedback and positive assessment of our paper. We address each point raised in detail below:
>
> **1. Checkpoint Requirements and Impact of Trajectory Length (T):**
> We acknowledge the concern regarding the practicality of frequent checkpoints and address this limitation in Appendix C, where we provide an analysis on selecting fewer (down sampling) checkpoints. Specifically, we demonstrate that the relevancy score remains consistent when we reduce the frequency of checkpoints (e.g., by skipping epochs). However, we also observe that as the down sampling rate increases, the relevancy score tends to approach 1, reducing the metric's sensitivity to sample-specific learning dynamics. To balance computational efficiency with accuracy, we suggest using every second or third checkpoint, which retains sufficient granularity for differentiating learning patterns while reducing the computational burden. The trajectory length T similarly impacts the metric, with shorter trajectories providing less detailed information but offering a feasible option for lower-resource environments.
>
> **2.	Generalization Across Tasks Beyond Classification:**
> The current work focuses on supervised classification tasks to establish the foundational utility of the relevancy metric. While we agree that exploring its application in regression or sequential tasks could significantly enhance its impact, extending the metric to such tasks involves additional considerations that are beyond the current paper’s scope. We are excited by the potential to adapt and test the relevancy metric in these domains as part of future research. This extension would require validating the metric's adaptability to different learning dynamics characteristic of non-classification tasks, which we believe will further strengthen the generalizability and utility of our approach.

---

> > ### Comment · Reviewer_jfAD · 2024-11-26
> > **Official Comment**
> >
> > I thank the authors for their responses. After checking the rebuttal and the reviews from other reviewers, I decided to lower my score unless the author addressed other reviewers' concerns.

---

> > > ### Author Response · Authors · 2024-11-26
> > >
> > > We thank the reviewer for their thoughtful feedback and appreciate their consideration of other reviews. We would like to kindly request the reviewer to revisit our rebuttal, where we addressed the concerns raised both by you and others.

---

### Meta-Review · Area_Chair_dpVR · 2024-12-17

**Metareview:**

Summary: This paper proposes a new metric, Relevancy, to evaluate the influence of individual training samples on model predictions. The metric is based on the trajectory of training loss value and requires the storage of model checkpoints across different epochs. The authors show applications of the proposed metric on the tasks such as coreset generation, memorization analysis, and mislabeled sample detection.

Strengths:
1. Evaluating the influence of individual training samples on model predictions is an important research question.
2. The paper is well-written and easy to read.
3. Relevancy is simple, efficient, and computation-friendly.

Weaknesses:
1. Relevancy depends on the loss trajectory during the training phase, which requires the storage of model checkpoints across epochs. This limits the application of Relevancy to broader settings where checkpoints might be unavailable (e.g., too large to be stored).
2. Reviewers are concerns about what specific type of impact the paper aims to quantify using the relevancy metric.
3. Reviewers have concerns about the missing of experiments comprising with other baselines, such as influence functions.
4. After rebuttal, reviewers are still concerns about the lack of robustness/reliability and rank-order consistency at large epochs.

Three out of four reviewers evaluate the paper with a score of 3, while the only positive reviewer lower his/her score after the rebuttal. Therefore, AC would recommend to reject the paper. AC would encourage the authors to take the above-mentioned weaknesses into consideration in the next revision of this paper.

**Additional Comments On Reviewer Discussion:**

In the first round of review, three out of four reviewers evaluate the paper with a score of 3. After the rebuttal, the three reviewers express their concerns on the rebuttal and are not willing to change their scores, while the only positive reviewer lowers his/her score. Given that the rebuttal is not very convincing to the reviewers and consistent negative scores, AC would follow reviewers' opinion to reject the paper.

---

### Decision · Program_Chairs · 2025-01-22

Reject